# Histopathologic, Genetic and Molecular Characterization of Endometrial Cancer Racial Disparity

**DOI:** 10.3390/cancers13081900

**Published:** 2021-04-15

**Authors:** Pouya Javadian, Christina Washington, Shylet Mukasa, Doris Mangiaracina Benbrook

**Affiliations:** 1Division of Gynecologic Oncology, Department of Obstetrics and Gynecology, Stephenson Cancer Center, University of Oklahoma Health Sciences Center, Oklahoma City, OK 73104, USA; CHRISTINA-WASHINGTON@ouhsc.edu; 2Arkansas College of Osteopathic Medicine, Fort Smith, AR 72916, USA; smukasa@arcomedu.org

**Keywords:** endometrial cancer, racial disparity, histopathology, molecular profile, targeted agents

## Abstract

**Simple Summary:**

Black patients are diagnosed and die earlier of endometrial cancer in comparison with their White counterparts. Factors that have been implicated in this racial disparity, such as socioeconomic status, increased frequencies of more aggressive tumor histology, and comorbid conditions, do not account for all of the disparity. Molecular defects in the endometrial tumors likely also contribute to the more aggressive tumor biology and the patient disparities. In this study, we reviewed the published data of molecular characteristics of endometrial cancer in different races. The majority of the publications compare Black and White patients, and identify molecules and pathways that can be targeted with existing drugs. These findings encourage molecular profile studies comparing additional races and ethnicities, and development of race-specific treatments.

**Abstract:**

In contrast to the decline in incidence and mortality of most other cancers, these rates are rising for endometrial cancer. Black women with endometrial cancer have earlier diagnosis, more aggressive histology, advanced stage and worse outcomes compared with their White counterparts. Socioeconomic status, a higher incidence of aggressive histology, and comorbid conditions are known factors leading to racial disparity in patients with endometrial cancer; nevertheless, they do not account for the entire racial disparity; which emphasizes the roles of molecular, histopathological and genetic factors. We performed a comprehensive review of all published scientific literature up to January 2021 reporting histopathologic, genetic and molecular factors associated with racial disparities in patients with endometrial cancer. The interactions and pathways of molecules reported to have significant differential expression in endometrial cancers from Black and White patients were identified with Ingenuity Pathway Analysis. The majority of studies compared Black and White patients; however, limited data are available for other racial and ethnic groups. Reported differences that could account for the worse survival of Black endometrial cancer patients include more aggressive histopathologies and molecular alterations, including upregulation of molecules driving cell cycle progression, and p53 and HER2/NEU signaling. Several of these molecules are targeted by existing pharmaceuticals. These findings encourage further study and the development of race-specific treatment strategies.

## 1. Introduction

Racial disparity remains the major factor affecting survival in patients with endometrial cancer in the US. Although the incidence of endometrial cancer has been reduced by 30% among Black patients, the mortality of endometrial cancer is 2.5 times higher in comparison with White patients [1]. This represents one of the largest racial disparities in mortality among common cancers [2]. The incidence of uterine cancers has continued to rise over time, while poorer survival continues to be reported among Black patients [3], even after adjusting for hysterectomy [4]. The contributions of socioeconomic factors to endometrial cancer racial disparities have been reviewed elsewhere [5,6].

A tumor’s genetic profile has been shown to play an important role in targeted therapy and chemotherapy regimens. The landmark paper by the Cancer Genome Atlas (TCGA) in 2013 described four molecular classes of endometrial cancer based on comprehensive and integrated molecular profiling [7]. This molecular profiling showed prominent prognostic advantages and opened the new era of clinic-pathological risk assessment in endometrial cancer patients. Based on the TCGA model, endometrial cancer is classified into four different molecular subgroups including: (1) the group with DNA polymerase epsilon, catalytic subunit mutated (*POLE*) mutations and its corresponding ‘ultramutated’ phenotype (POLEmut) (pathogenic mutations in the exonuclease domain of POLE in this subgroup are associated with favorable outcomes even with high-grade tumors [8,9,10]); (2) the microsatellite instability (MSI) group, which shows an intermediate prognosis; (3) copy-number low (CN low) and (4) copy-number high (CN high), which are substantially related to high-grade serous cancers and poor prognosis. The first three subgroups are known to have few copy number alterations and few *TP53* mutations. It has been shown that they have more frequent mutations in Phosphatase and Tensin Homolog (*PTEN*), Catenin Beta 1 (*CTNNB1*), AT-Rich Interaction Domain 1A (*ARID1A*), *ARID5B*, and *KRAS*. Comparatively, Group 4, comprising 94% of uterine serous carcinomas (USCs), which represent the most aggressive histologic type, and 12% of endometrial adenocarcinomas, demonstrated extensive copy number alterations of the oncogenes *MYC*, Erb-B2 Receptor Tyrosine Kinase 2 (*ERBB2*) and cyclin E1 (*CCNE1*), and frequent *TP53* mutations [7].

The randomized Adjuvant Chemoradiotherapy Versus Radiotherapy Alone in Women with High-Risk Endometrial Cancer (PORTEC-3) trial compared adjuvant radiation and chemotherapy with radiation alone in patients with high-risk endometrial cancer. They measured p53 and mismatch repair (MMR) proteins with *POLE* gene sequencing to categorize patients into four molecular subgroups (similar to TCGA) and demonstrated differences in treatment response across groups [11]. They revealed that in those with p53 mutations, combined adjuvant chemotherapy and radiation reduced the 5-year recurrence risk by 50% in comparison with those that received radiation alone. These studies highlight the potential of utilizing molecular profiling to develop individualized treatments that could be applied to address racial disparities.

In this review, we tried to investigate the current body of literature to identify molecular, biological, histopathologic and genetic factors associated with endometrial cancer disparities and identify potential pharmaceutical interventions.

## 2. Material and Methods

A literature review was performed using PubMed, Medline and the Cochrane Library. We included all the English literature available up to January 2021. We used specific search terms including endometrial cancer, uterus cancer, endometrial, uterine, proteomic profile, proteomic analysis, molecular profile, histology, genetics and genome. These were combined with the terms race or racial or African or Native or Caucasian or White or Asian or Indian or disparity or disparities. We further added more manuscripts which were pointed out in the bibliographies of the studies. We excluded those with duplicate publications, publications where outcomes of interest were not assessed and publications without full text. We further excluded duplicate studies using EndNote software, version X7.0 (Thomson Reuters, New York, NY, USA). Each author participated in the review process by screening the manuscripts for the title and abstract. Those of interest were included in the study by obtaining their full texts. Finally, 93 studies were included in this review, as shown in Figure 1.

The literature was also searched using the names of molecules reported to be differentially expressed in endometrial cancer specimens from Black compared with White patients in search of the molecule’s mechanisms of action and for pharmaceuticals that target these molecules.

The interactions and pathways of molecules identified reported to be differentially present in endometrial cancer specimens in Black compared with White patients were collectively evaluated using Ingenuity Pathway Analysis Software.

## 3. Results

### 3.1. Histology, Pathogenic Types and Endometrial Disparity

Endometrial cancers are categorized into different histologic subtypes including: endometrioid (75–80% of patients), USC (<10% of patients), clear cell (<5% of patients) and carcinosarcoma (<5% of patients) [3]. In the early 1980s, Bokhman suggested two pathogenic types of endometrial cancer based on their differences in histology and clinical outcomes. Type I tumors comprise the large majority of endometrial cancers, develop from and are associated with atypical glandular hyperplasia, are related to unopposed estrogen stimulation and are often preceded by endometrial hyperplasia. In a California cancer registry study, White patients were more likely to present with Type I endometrial cancer of Stage 1 disease compared with Black patients (45.8% vs. 34.9%, *p* = 0.0001) [12].

Type II histological subtypes include predominantly USC, and also clear cell, carcinosarcoma and high-grade endometroid cancer and mixed (typically endometrioid and a high-grade nonendometrioid pattern) [13]. In multiple studies, Black endometrial cancer patients had higher incidences of these aggressive histologic subtypes in comparison with non-Hispanic White patients [12,14,15,16,17,18,19]. Overall, Black patients have the highest incidence of these most aggressive, nonendometrioid histologies of endometrial cancer in the US population (Figure 2) [15].

The etiology of Type II tumors is less clear, as their development is not associated with estrogen exposure and they arise in atrophic endometrium from intraepithelial carcinoma in multiparous, nonobese, mostly older patients. Genetic and environmental factors have been implicated in their development [20]. In general, they are less well differentiated and have worse prognoses than Type I tumors, and they are responsible for a disproportionate number of endometrial cancer deaths. The increased incidence of Type II compared with Type I tumors is consistent with the worse 5-year disease-specific survival of Black (51–57%), compared with White (65–67%), Hispanic (64–69%) and Asian (67–72%) endometrial cancer patients in a California registry study (*p* < 0.0001) [12,21]. In a Florida registry study, Black endometrial cancer patients had a higher incidence of Type II tumors (57.6%) compared with White (35.6%), Hispanic (37.7%) and Asian (43.0%) endometrial cancer patients, and a 24% high risk of death due to endometrial cancer compared with white patients [21]. A national population-based study of 35,850 young women (<50 years of age) with endometrial cancer found that the young Black patients (3903) had higher mortality compared with the young White patients [19]. Histology alone, however, does not account for the worse survival of Black endometrial cancer patients. A study of data from the Surveillance, Epidemiology and End Results (SEER) program showed that Black in comparison with White patients had worse survival in every histologic category, stratified by stage, grade and age [16,22]. This study also found that rare aggressive tumor types accounted for a higher percentage of the deaths in Black (53%) compared with White (36%) patients.

### 3.2. Hereditary Cancer and Genetic Predisposition

Short tandem repeats (STRs), also called microsatellites, are short repeated DNA sequences present throughout the genome. The repetitive nature of microsatellite DNA sequences causes increased susceptibility to mutation during DNA replication [23], with a mutation frequency range of 10^−6^ to 10^−2^ per generation [24]. DNA mismatch repair (MMR) is a DNA repair pathway that proofreads newly replicated DNA and repairs mutations associated with replication of microsatellites [25]. Changes in the genes responsible for this important pathway could lead to Lynch syndrome; this can happen due to germline mutation in mismatch repair genes (*MLH1*, *MSH2*, *MSH6* or *PMS2*) or germline deletion of the 3′ portion of the Epithelial Cell Adhesion Molecule (*EPCAM*) gene [26]. Moreover, epigenetic alterations, such as DNA methylation in the gene promotor region, can suppress transcription and interfere with the expression of MMR genes [27]. Women with Lynch syndrome have an increased risk of developing endometrial cancer, and at higher and faster rates than their risk for colorectal cancer [28,29]. Currently, there is no evidence that race and ethnicity are associated with an increased frequency of Lynch syndrome. A study of a sample population of 62 Black and 78 White patients diagnosed with Stage III/IV endometrial adenocarcinomas found no racial disparities in the frequency of MSI (Whites, 16%; Blacks, 13%), in the subset with endometrioid histology (Whites, 20%; Blacks, 22%) [30]. Analysis of the TCGA dataset revealed that patients of Asian descent have a higher frequency of somatic mutations of MMR genes in contrast with White or Black groups of patients. Missense mutation is the most prevalent type of mutation in the Asian population and the MMR-associated gene, *PMS2*, was identified as most significant mutated gene among all other genes in the TCGA dataset (Fisher’s exact test; *p* = 0.0036) [10].

### 3.3. Global Profiling to Identify Racial Differences in mRNA Transcripts and Proteins

Publications of global profiling efforts to identify mRNA and proteins that could contribute to racial disparities in endometrial cancer are limited to two studies comparing endometrial cancer specimens in Black and White patients. The molecules identified in these studies are summarized in Table 1. In 2017, Bateman et al. reported on a discovery set of endometrioid endometrial cancer samples from 17 Black and 13 White patients, the results of which were validated using the independent TCGA dataset of endometrioid endometrial cancer patients from 49 Black and 216 White patients [31]. For the discovery set, laser capture was used to specifically harvest cancer cells for isolation of RNA, which was evaluated by whole-transcriptome sequencing, and protein, which was evaluated by liquid chromatography-tandem mass spectrometry (LC-MS/MS). The proteomic analysis identified 94 proteins as being altered between the Black and White groups. Ten of these candidates were reduced in expression at both the RNA and protein levels in cancers from Black compared with White patients (Table 1) [31].

Interestingly, there were no candidates identified with increased expression in Black patients. There were 587 mRNA transcripts that also exhibited altered protein levels, and 89 were validated to be concordant with the TCGA data (Table 1). Three of the ten candidates with altered mRNA and protein in the discovery set (EEF2, JUP and MYO1C) were among these 89. EEF2 was further validated to be decreased in Black endometrial cancer patients (irrespective of histology type) in reverse phase protein array (RPPA) TCGA data. Molecular pathway analysis of the 89 RNA transcripts and 94 proteins altered between groups identified that the Black patients exhibited activation of pathways involved in tumor cell viability, nucleic acid metabolism and counteraction of cell death, and inhibition of pathways involved in regulating viral infection. Multivariate analysis of the 89 RNA transcripts validated in the TCGA data found significant associations with progression-free survival (PFS) for 10 transcripts in White patients, nine transcripts in Black patients, and two transcripts in both White and Black patients (Table 1). A higher prevalence of copy number-high, Cluster 4 and mitotic subtypes was observed in Black compared with White patients with endometrioid histology, and found that specimens from Black patients exhibited higher transcript levels for cell cycle regulatory proteins, PLK1 and BIRC7 (Table 1) [32]. Pathways identified in both of these studies are cell cycle and inhibition of cell death [31,32]. After adjustment of the data based on tumor characteristics, several genes (e.g., serpin family A member 4 (*SERPINA4)* and blocked early in transport 1 homolog (BET1L)) were still associated with better PFS in White patients only. SERPINA4, a prominent anti-angiogenesis factor, plays an important role in tumor growth inhibition, especially in colorectal cancer. BET1L, a protein receptor that regulates Golgi vesicular membrane trafficking, is associated with an increased chance of uterine fibroids. It has been shown that a change in a single nucleotide polymorphism of the *BET1L* gene is related to a lower risk of uterine fibroids in European Americans [54]. DNA Ligase 3 (*LIG3*) was related to improved PFS for White patients but poor PFS for Black patients. LIG3’s expression can increase genome instability and cancer risk, which could lead to greater chance of poor outcomes for Black patients compared with White. Contrarily, family with sequence similarity 228 (*FAM228B*), was found to be associated with better PFS in Black patients only. [55]

A limitation of these studies is that they did not match the patients in the different racial groups for factors that could affect the aggressiveness of their cancer and their treatment outcomes, such as age, tumor stage and BMI, although they did focus on one histology: endometrioid. Several studies which matched specimens in the different racial groups found no significant transcript differences. No global gene expression differences were found in a microarray study of endometrial cancer specimens from Black women matched in one-to-one, two-to-one and three-to-one ratios with White women for stage, grade and histologic subtype [56]. Similarly, no global gene expression differences were found in a later microarray study of laser micro-dissected endometrial cancer specimens from 25 Black and 25 White patients matched for histology (endometrioid and serous), grade (1A through IVB) and grade (1–3) [57].

### 3.4. Racial Differences in MicroRNAs

A comparison of laser micro-dissected unmatched Stage I endometrioid endometrial cancer specimens from 41 White and 9 Black patients found no global differences; however, a subset analysis of stage- and histology-matched specimens from nine White and nine Black patients identified alterations in 18 microRNAs (miRNAs) [58]. The miRNA-337-3p was validated by quantitative polymerase chain reaction to be differentially expressed in endometrial cancer specimens but not in normal endometrium from an independent set of 23 White and 24 Black women. Targets of miRNA-337-3p include STAT3, which is upregulated by multiple miRNAs, long noncoding RNAs and circular RNAs, and drives endometrial cancer development, progression and metastasis [59,60,61,62].

Pathway analysis of the SEER program database from 2000 to 2015 in patients classified as European-American or African-American was performed to determine the clinical significance and biologic functions of the differentially expressed miRNAs related to cancer type [22]. They reported that differentially expressed miRNAs were associated with cancer drug resistance pathways through cancer drug efflux for all cancer types including uterine. In order to further elaborate cancer drug resistance, they analyzed DNA methylation and gene expression in different cancer types. The result revealed that 157 genes with dysregulated methylation were associated with uterine/endometrial cancer in African-American patients. They concluded that altered DNA methylation, along with increased microRNA expression levels in African-American patients, is associated with cancer drug resistance in this patient population [22].

### 3.5. A Targeted Approach to Identifying Racial Differences in Specific Proteins

Another approach to identifying the biological factors responsible for racial disparities are targeted studies that evaluate specific molecular candidates for differential expression and associations with survival between racial groups (Table 1). The most well-characterized candidate is the tumor suppressor *TP53* gene, which is the most commonly mutated gene in all cancer types. *TP53* mutations typically are either p53 null, which causes p53 protein truncation or loss of expression, and missense point mutations, which cause overexpression and oncogenic gain-of-function of the p53 protein. In patients with endometrial cancer, p53 overexpression has been shown to be related to aggressive tumor characteristics, including lymph node metastases, poor differentiation and deep myometrial invasion [63,64]. Several studies found a higher expression of p53 protein in endometrial cancer specimens from Black in comparison with White patients [65,66]. In Black patients with endometrial cancer, a higher frequency of p53 overexpression increased the chance of Type II cancer in this patient population [66]. The rate of p53 overexpression in patients with Type I cancer was not different between the two ethnic groups [66]. The rate of p53 overexpression observed in endometrial cancers from White patients was similar to the rate observed in Japanese patients reported by the same research group [67,68].

Loss of tumor suppressor PTEN protein expression is common in endometrial glands associated with hyperplasia and cancer [69]. The *PTEN* tumor suppressor gene encodes a phosphatase that inhibits proliferation by opposing the activity of oncogenic kinases, such as those involved in the AKT/PIC-3A pathway. Inactivating *PTEN* gene mutations have been reported in 55% of precancers and 83% of endometrioid endometrial cancers; however, validation studies found PTEN loss to have limited prognostic value for the progression of endometrial hyperplasia or endometrial intraepithelial neoplasia (EIN) pre-cancers to cancer [70,71]. In a DNA sequencing study of 78 White and 62 Black unmatched Stage III/IV frozen endometrial adenocarcinoma specimens, specimens from Black patients had significantly fewer PTEN mutations (5%) in comparison with specimens from White patients (22%) [30]. Cancer cells with PTEN expression have been shown to have a more favorable response to chemotherapy; however, overexpression of miR-130a may upregulate the proliferation of chemo-resistant cancer cells by inhibiting PTEN expression and, consequently, activation of the PI3 K/AKT signal pathway that increases BCL-2 expression to decrease tumor cell apoptosis [72].

Black individuals were noted to have higher frequencies of germline single nucleotide polymorphisms (SNPs) in *PTEN* genes compared with matched White individuals with European ancestry [73]. A strong co-existence of *PIK3CA* and *PTEN* mutations was observed in the British South Asian group in comparison with the British White group. The correlation between *PIK3CA* and *PTEN* has also been reported previously in endometrial cancer in a cohort of cases from Japan [74]. *PTEN* mutations are associated with favorable prognosis in endometrial cancer patients [75]. Cancer cells with PTEN expression have been shown to have a more favorable response to chemotherapy; however, overexpression of miR-130a may upregulate the proliferation of chemo-resistant cancer cells by inhibiting PTEN expression and, consequently, activation of the PI3 K/AKT signal pathway that increases BCL-2 expression to decrease tumor cell apoptosis.

Targeted studies of vascular endothelial growth factor (VEGF), Ki-67 and hypoxia inducible factor-1α (HIF-1α) expression found no differences in endometrial cancer specimens from Black and White patients [66].

Morrison et al. reported that HER2 is an important prognostic factor in high-risk endometrial cancer. They showed a trend for a higher rate of HER2 expression (33%) among Black patients in contrast to White patients (13%) [76]. Furthermore, it has been reported that HER2 is more frequently upregulated in endometrial cancers from Black women [18]. A study of USC specimens from 10 Black and 17 White patients (not matched) demonstrated higher HER2/neu levels in Black compared with White patients, and the association of high HER2/neu expression with worse USC patient outcomes in both races [77].

Phosphoserine phosphatase like (PSPHL) was initially identified as an upregulated gene in the fibroblasts of patients with Fanconi anemia. Higher *PSPHL* expression levels have been reported in Black patients with different types of cancer compared with White patients. [78,79]. Although global differences in the transcripts expressed by genes were not evident between Black and White endometrial cancer patients, a subset of significantly differentially expressed PSPHL transcripts was observed, which was validated by quantitative real-time PCR. PSPHL was found to be prominently expressed in endometrial cancer specimens from Black women, and the group cloned an additional splice variant that was also preferentially expressed in Black women [80].

High insulin growth factor 2 (IGF2) expression is associated with chemotherapy resistance and is a candidate factor that could contribute to the worse treatment responses of Black compared with White patients with USC. Immunohistochemical analysis of 103 USC specimens found a higher IGF2 expression in Black compared with White patients, and a race-stratified multivariate analysis found that high IGF2 expression in the epithelial component of the tumors more than doubled the risk of death in Black women [81].

### 3.6. Pathway Analysis of Racial Differences in mRNA and Proteins

We conducted an Ingenuity analysis to identify associations between proteins exhibiting altered expression in endometrial cancer specimens from Black compared with White patients and that were also prognostically significant in Black patients (Figure 3). The analysis demonstrated that p53 transcription factor increases the expression of PTEN, HER2/NEU and IGF2, proteins which also are known to interact with each other. The missense p53 mutations that can cause p53 protein overexpression, however, often alter the profiles of the genes regulated by p53. HER2/NEU is an indirect upstream regulator of IGF2, LIG3 and PBX1, and IGF2 regulates PTEN expression.

Analysis of all of the molecules listed in Table 1 confirmed the significant association of molecules differentially expressed in Blacks with the cell cycle regulation pathway and showed pathway associations among the cell cycle regulatory proteins along with EEF2, HER2/NEU, FGFR3 MCM2/7 and BRCA2 (Figure 4).

### 3.7. Race-Based Treatment and Outcome

A growing body of evidence suggests that targeted therapies have an improved capacity for improving cancer patient outcomes with less toxicity compared with chemotherapy [82,83]. Despite encouraging outcomes for these new therapies, the participation of minorities in clinical trials remains low. Specifically, Black patients are underrepresented in federally-funded clinical trials [12]. This represents a highly significant issue, especially given the fact that Black women experience worse outcomes compared with White women with the same disease. A gynecologic oncology group study of patients receiving treatment for advanced endometrial cancer in clinical trials found that 169 Black patients experienced 20% worse survival compared with 982 White patients, even though the participants received uniform care [84]. The disparity remained when the researchers controlled for socioeconomic status and tumor stage, grade and histology. Further investigation demonstrated that Black patients experienced lesser responses to chemotherapy regimens consistently across several clinical studies. Taken together, these findings suggest that there are potential molecular and biological components contributing to the observed racial disparity, thus emphasizing the promise of new molecularly-targeted therapies in Black women.

New therapies require evaluation in clinical trials to determine their effectiveness as potential treatment options. Phase I clinical trials allow researchers the ability to determine the maximum tolerated dose and the optimal administration of new drugs in patients. There is a lack of minority representation in Phase I clinical trials, which may lead to the disparities seen in cancer outcomes [85].

### 3.8. Candidate Drugs for Race-Specific Treatment of Black Endometrial Cancer Patients

Molecularly targeted drugs offer the opportunity to have a greater impact on endometrial cancer patients belonging to races with known alterations in those targets. Of the race-specific proteins listed in Table 2, p53 and HER2/NEU (*ERBB2*) have molecularly targeted agents in development or approved for cancer treatment. The higher expression of p53 protein in endometrial cancer specimens from Black patients may be due to the higher *TP53* gene mutation rate associated with USC, which occurs at a higher rate compared with other histologies in Black women. Multiple chemical and biologic drugs have been developed against mutant p53; however, none are currently approved [86,87]. There are multiple drugs targeting HER2/NEU, including the humanized monoclonal antibody, trastuzuab. A randomized Phase 2 trial of adding trastuzumab to carboplatin-paclitaxel showed that this combination was well tolerated and improved PFS by 4.6 months over chemotherapy alone in patients with advanced-stage or recurrent USCs that overexpressed HER2/NEU [88]. This improvement was >8 months for patients with advanced-stage disease undergoing primary treatment and >3 months for patients with recurrent disease. Follow-up analysis showed that overall survival was increased by >4 months in the trastuzumab arm, with no increase in toxicity [89]. Currently, there are two clinical trials of HER2/NEU inhibitors for endometrial cancer listed on clinicaltrials.gov (access date: 10 January 2021).

Most molecules listed in Table 1 do not have targeted drugs; however, there are several molecules which can be targeted by specific drugs to provide potential race-specific treatment. As discussed previously, the most represented pathway among these genes is cell cycle regulation, for which there are multiple drug options. Currently, there are multiple drugs targeting cyclins and cyclin-dependent kinases (CDKs) in clinical trials that provide potential for inhibiting cyclin E overexpression in endometrial cancers of Black patients [36,37]. The elevated CDKN2A transcript can be alternatively spliced to produce p14 or p16 proteins. The p16 protein is characteristically overexpressed in senescent and cancer cells [91,92]. There are multiple drugs, termed senoltyics, that have been shown to selectively kill senescent cells and are being studied as anti-cancer agents [38,39]. While there are drugs specifically targeted at CDC7, the CDC7 transcript is decreased in Blacks and thus is not a rational target for Black race-specific endometrial cancer treatment. PIK3C, an upstream regulator of cell cycle regulatory proteins [93], has been effectively targeted with inhibitors in breast cancer [40,41,42,43].

Several of the molecules listed in Table 1 have molecularly targeted inhibitors in various phases of development for a variety of cancers. The PLK1 protein has been targeted by multiple biologic and chemical drugs [48,49,50,51,52,53]. Multiple drugs that co-inhibit FGFR3 along with other plasma membrane receptors are being studied as cancer therapeutics [44,45].

Other molecules listed in Table 1 can be inhibited indirectly by known drugs. The expression of PBX1 is decreased by camptothecin, which is a topoisomerase poison used to treat leukemia. Multiple formulations of camptothecin analogs are currently in clinical trials for a variety of cancers (clinicaltrials.gov (access date: 10 January 2021)). EEF2 can be inactivated by a biologic drug called MDNA55, which is a circularly permuted version of interleukin-4 (IL-4) conjugated to *Pseudomonas* Exotoxin A (PE). Upon binding to the IL-4 receptor, the drug is internalized in the cell, where the PE is released and causes ADP-ribosylation of EEF2 on ribosomes [35]. There are also other biologics and immunotoxins that are conjugated with PE and provide additional candidate drugs for race-specific treatment of endometrial cancer [94].

The mutational profiles of tumors have also been shown to affect sensitivities to specific cancer therapies. For example, in a Phase 2 clinical trial of 86 patients with MMR deficiencies across different solid tumors, including endometrial cancer, pembrolizumab inhibition of PD-1 resulted in a 77% disease control rate [95]. In this study, genomic signatures predicted tumor phenotype and treatment responses independently of the anatomic site of the disease or the histology of the tumor. Therefore, PD1 blockade may provide improved outcomes in endometrial cancer patients of Asian descent, based on the higher frequency of somatic mutations of MMR genes in Asian endometrial cancer patients.

## 4. Discussion

Our review of the literature identified molecules and molecular pathways that are present at significantly different levels in endometrial cancer specimens from Black and White patients, and encourages controlled studies of other races and ethnicities. Several of these molecules and their pathways can be targeted by existing pharmaceuticals, which have potential for use in race-specific treatment strategies. Recent studies have revealed the potential for improved patient management by enabling tumors to be classified into molecularly categorized subgroups, in addition to the four TCGA groups, with significantly different tumor responses to treatment patient prognoses. These results encourage further elucidation of the molecular pathways and genetic characteristics of endometrial cancers for all race and ethnic groups exhibiting disparities. This information can be used to develop better predictive models of cancer progression and novel therapeutic approaches.

## Figures and Tables

**Figure 1 cancers-13-01900-f001:**
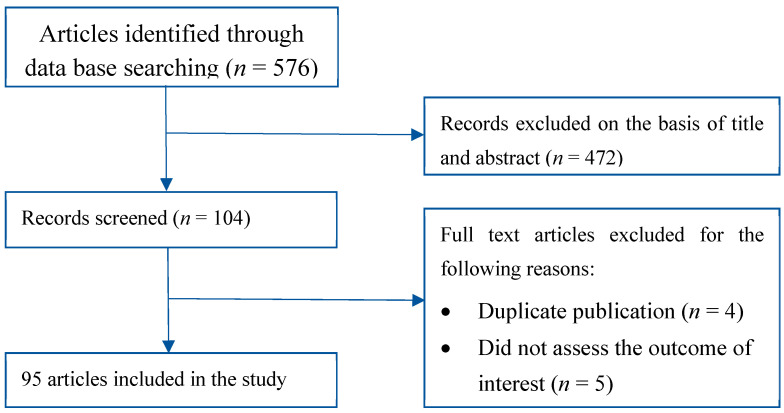
Literature search and review flow chart for selection of studies.

**Figure 2 cancers-13-01900-f002:**
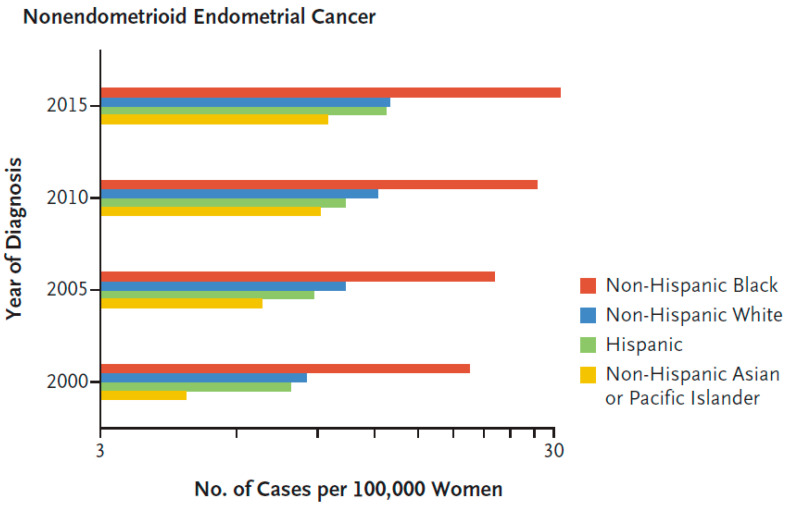
Age-adjusted and hysterectomy-corrected incidence of nonendometrioid endometrial cancer according to race [15]. Reproduced with permission from Massachusetts Medical Society.

**Figure 3 cancers-13-01900-f003:**
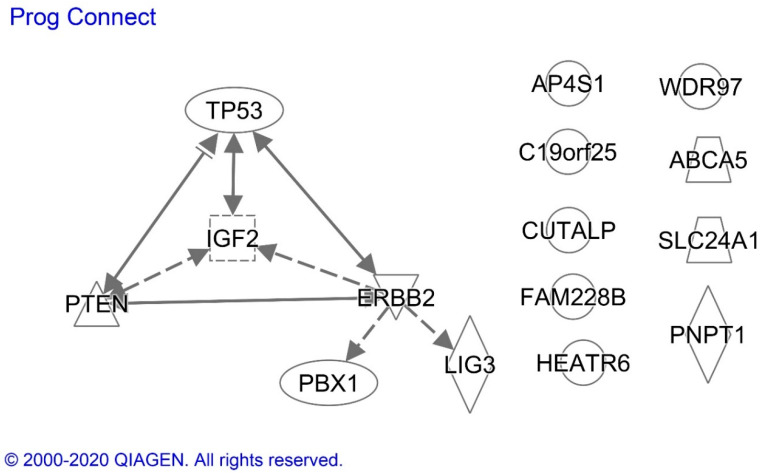
Ingenuity analysis of prognostically significant proteins with altered expression in endometrioid endometrial cancer specimens from Black compared with White patients. Solid and dotted lines indicate the direct and indirect effects, respectively, of the indicated protein on the protein to which the arrowhead is pointing. ERBB2 = HER2/NEU. Proteins without arrows had no evidence of a connection with the other proteins in the current scientific literature.

**Figure 4 cancers-13-01900-f004:**
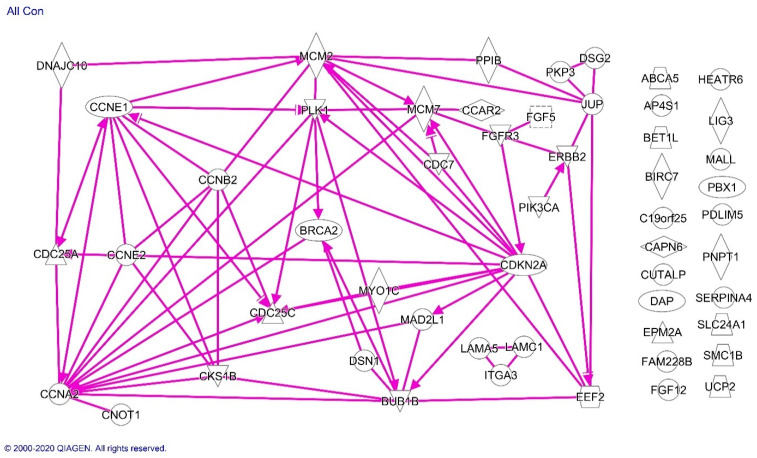
Ingenuity analysis of proteins differentially expressed in endometrioid endometrial cancer specimens from Black compared with White patients. Solid lines indicate direct effects of the indicated protein on the protein to which the arrowhead is pointing. Proteins without arrows had no evidence of a connection with the other proteins in the current scientific literature.

**Table 1 cancers-13-01900-t001:** Molecules significantly associated with race and survival.

Difference	Name (Gene Symbol)	Survival Association	Reference
RNA Increased in Black Patients	DSN1 homolog (*DSN1*) ^a^	PFS in Black and White	[31]
	PBX homeobox 1 (*PBX1*) ^a,d^	PFS in Black ^b^	[31]
	EPM2A laforin glucan phosphatase (*EPM2A*)^a^	PFS and OS in White ^b^	[31]
	DNA Ligase 3 (*LIG3*) ^a^	PFS worse for Black and improved for White	[31]
	Bet1 Golgi vesicular membrane trafficking protein like (*BET1L*) ^a^	PFS in White	[31]
	HEAT repeat containing 6 (*HEATR6*) ^a^	PFS in Black	[31]
	Family with sequence similarity 228 member B (*FAM228B*) ^a^	PFS in Black ^b^	[31]
	Adaptor-related protein complex 4 r1 subunit (*ARCA5*) ^a^	PFS in Black	[31]
	PSMD5 antisense RNA 1 (head to head) (*PSMD5-AS1*) ^a^	PFS in Black ^b^	[31]
	Polyribonucleotide nucleotidyltransferase 1 (*PNPT1*) ^a^	PFS in Black	[31]
	Adaptor-related protein complex 4 r1 subunit (*AP4S1*) ^a^	PFS in Black ^b^	[31]
	Cyclin E1 (*CCNE1*) ^a,c,d^	Not determined	[32]
	Cyclin dependent kinase inhibitor 2A/P16 (*CDKN2A*) ^c,d^	Not determined	[32]
	Cell division cycle 25C (*CDC25C*) ^c^	Not determined	[32]
	Cyclin B2 (*CCNB2*) ^c^	Not determined	[32]
	Mitotic checkpoint serine/threonine kinase B (*BUB1B*) ^c^	Not determined	[32]
	Minichromosome Maintenance complex component 7 (*MCM7*) ^c^	Not determined	[32]
	Polo like kinase 1 (*PLK1*) ^c,d^	Not determined	[32]
	Minichromosome maintenance complex component 2 (*MCM2*) ^c^	Not determined	[32]
	Baculoviral inhibitor of apoptosis repeat containing 7 (*BIRC7*) ^c^	Not determined	[32]
	Laminin subunit gamma 1 (*LAMC1*) ^c^	Not determined	[32]
	CDC28 protein kinase regulatory subunit 1B (*CKS1B*) ^c^	Not determined	[32]
	Laminin subunit 5 (*LAMA5*) ^c^	Not determined	[32]
	Wnt family member 7A (*WNT7A*) ^c^	Not determined	[32]
	Fibroblast growth factor 12 (*FGF12*) ^c^	Not determined	[32]
	Fibroblast growth factor 5 (*FGF5*) ^c^	Not determined	[32]
	FGF receptor 3 (*FGFR3*) ^c,d^	Not determined	[32]
	Erb-B2 receptor tyrosine kinase 2 (*ERBB2*) ^c,d^	Not determined	[32]
	Breast cancer 2 DNA repair associated (*BRCA2*) ^c^	Not determined	[32]
	Phosphatidylinositol-4, 5-bisphosphate3-kinase catalytic subunit α (*PIK3CA*) ^c,d^	Not determined	[32]
RNA Decreased in Black Race	Wilms tumor 1 associated protein (*WTAP*) ^a^	PFS in White	[31]
	Death-associated protein (*DAP*) ^a^	PFS and OS in White ^b^	[31]
	Mal T-cell differentiation protein like (*MALL*) ^a^	PFS in White ^b^	[31]
	Integrin subunit a3 (*ITGA3*) ^a^	PFS and OS in White	[31]
	Uncoupling protein 2 (*UCP2*) ^a^	PFS in White ^b^	[31]
	Serpin family A member 4 (*SERPINA4)* ^a^	PFS and OS in White ^b^	[31]
	Calpain 6 (*CAPN6*) ^a^	PFS in White ^b^	[31]
	Solute carrier family 24 member 1 (*SLC24A1*) ^a^	PFS in Black	[31]
	Chromosome 19 open reading frame 25 (*C19orf25*) ^a^	PFS in Black	[31]
	WD repeat domain 97 (*WDR97*) ^a^	PFS in Black	[32]
	Cyclin A2 (*CCNA2*) ^c^	Not determined	[32]
	Mitotic arrest deficient 2 like 1 (*MAD2L1*) ^c^	Not determined	[32]
	Cyclin E1 (*CCNE2*) ^c^	Not determined	[32]
	Cell division cycle 25C (*CDC25A*) ^c^	Not determined	[32]
	Cell division cycle 7 (*CDC7*) ^c,d^	Not determined	[32]
	Structural maintenance of chromosomes 1B (*SMC1B*) ^c^	Not determined	[32]
RNA and Protein Decreased in Blacks	CCR4-NOT transcription complex subunit 1 (*CNOT1*)	Not determined	[31]
	Peptidylprolyl isomerase B (*PPIB*)	Not determined	[31]
	Cell cycle and apoptosis regulator 2 (*CCAR2*)	Not determined	[31]
	Eukaryotic translation elongation factor 2 (*EEF2*)^d^	Not determined	[31]
	Junction plakoglobin (*JUP*)	Not determined	[31]
	PDZ and LIM domain 5 (*PDLIM5*)	Not determined	[31]
	Myosin IC (*MYO1C*)	Not determined	[31]
	Desmoglein 2 (*DSG2*)	Not determined	[31]
	Plakophilin 3 (*PKP3*)	Not determined	[31]
	DnaJ heat shock protein family (Hsp40) member C10 (*DNAJC10*)	Not determined	[31]

^a^ Validated in TCGA RPPA data. ^b^ Significant in multivariate COX analysis for stage, grade and myometrial invasion; Wald *p* < 0.05. ^c^ Mitotic subtype only. ^d^
**Actionable Target: *PBX1—***camptothecin [33,34]; ***EEF2—***MDNA55 [35]; ***CCNE1—***CDK2 and Akt inhibitors [36,37]; ***CDKN2A—***senolytics, PLK1- BI 2536, GSK461364, lipid-encapsulated anti-PLK1 siRNA TKM-080301, MK1496, onvansertib, rigosertib, TAK-960 and volasertib [38,39]; ***CDC7—***BMS-863233, LY3143921, NMS-1116354, TAK-93 [40,41,42,43]; ***FGFR3—***3D185, ASP5878, B701, crizotinib/pazopanib and debio [44,45]; ***ERBB2—***^68^GaNOTA-anti-HER2 VHH1, ^89^Zr-trastuzumab, ^99^m-Tc-NM-02, ^99^mTc-HPArk2, [^68^Ga] ABY-025, A166, ADCT-502, afatinib, afatinib/cetuximab, afatinib/osimertinib, afatinib/paclitaxel, allitinib and alpelisib/trastuzumab [46]; ***PIK3CA—***alpelisib, alpelisib/fulvestrant, alpelisib/trastuzumab emtansine, AZD8835, BAY1082439, bimiralisib, buparlisib, CLR457, copanlisib, dactolisib, GDC-0077, pictilisib, pilaralisib, PKI-179, PWT33597, PX-866, serabelisib, SF 1126, taselisib [40,41,42,43]; ***PLK1—***volasterib, fostamatinib, cytarabine, decitabine [47,48,49,50,51,52,53].

**Table 2 cancers-13-01900-t002:** Race-specific proteins as drug targets in endometrial cancer.

Protein	Method	Racial Difference	Result
p53 [90]	Immunohistochemistry in FFPE of all histology types	Increased in Blacks	Higher expression independently associated with worse survival in all patients
PTEN [30]	DNA mutations that decrease expression	Decreased in Blacks	Mutations are associated with more favorable outcomes
HER2/Neu [18]	Immunohistochemistry in FFPE of USC	Increased in Blacks	Higher expression associated with worse disease-related survival in all races
IGF2 [81]	Immunohistochemistry	Increased in Blacks	Higher expression associated with worse survival in all races and doubling of risk in Blacks with USC

FFPE: Formalin-fixed, paraffin-embedded.

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
