# Peer review of "Histopathologic, Genetic and Molecular Characterization of Endometrial Cancer Racial Disparity"

_cancers, 2021, doi:10.3390/cancers13081900_

Round 1

Reviewer 1 Report

This is a very nice review summarising the current status of racial disparity in endometrial cancer incidence. I don't have any major concerns, but there are  few things i would like clarifying.

1) Guttery et al. 2018 and Zhang et al.  2017 performed a comprehensive analysis of the TCGA UCEC dataset, and Lara et al. (2019) used the SEER dataset. They showed a number of key differences across all analytes. Why was these studies not included here?

2) Table 1 - I'm not sure whether the pathways with "not determined" are necessary since they don't seem to add anything.

3) I think it should be noted or discussed that although HER2 amplified ECs do indeed have poorer prognosis, there should be discussion around the marginal benefits Herceptin offers.

Author Response

  • Guttery et al. 2018 and Zhang et al. 2017 performed a comprehensive analysis of the TCGA UCEC dataset, and Lara et al. (2019) used the SEER dataset. They showed a number of key differences across all analytes. Why was these studies not included here?

Thank you for your recommendations regarding these important manuscripts, we have added these studies (10, 22 and 73) to the references list and cited them in different sections of our paper as it is colored in red.   The consequently newly added information on higher MMR deficiency in Asian compared to Black and White endometrial cancer patients inspired us to move the sentence “For example, in a Phase 2 clinical trial of 86 patients with MMR deficiencies across different solid tumors, including endometrial cancer, pembrolizumab inhibition of PD-1 resulted in a 77% disease control rate [96]. In this study, genomic signatures predicted tumor phenotype and treatment responses independently of the anatomic site of the disease or the histology of the tumor.” from the first paragraph under “3.7 Race Based Treatment and Outcome” to a new final paragraph under “3.8 Candidate Drugs for Race-Specific Treatment of Black Endometrial Cancer Patients”. And added the two sentences highlighted in red as shown below:

Mutational profiles of tumors have also been shown to affect sensitivities to specific cancer therapies. For example, in a Phase 2 clinical trial of 86 patients with MMR deficiencies across different solid tumors, including endometrial cancer, pembrolizumab inhibition of PD-1 resulted in a 77% disease control rate [96]. In this study, genomic signatures predicted tumor phenotype and treatment responses independently of the anatomic site of the disease or the histology of the tumor. Therefore, PD1 blockade may provide improved outcomes in endometrial cancer patients of Asian descent, based on the higher frequency of somatic mutations of MMR genes in Asian endometrial cancer patients.    

  • Table 1 - I'm not sure whether the pathways with "not determined" are necessary since they don't seem to add anything.

We understand the reviewer’s point, however we believe reporting “not determined pathways” would emphasize the importance of further investigations. 

  • I think it should be noted or discussed that although HER2amplified ECs do indeed have poorer prognosis, there should be discussion around the marginal benefits Herceptin offers.

We have discussed the importance of HER2 ECs in multiple sections of the manuscript and in order to further discuss the importance of Herceptin therapy in these patients we added the sentences in red to the first paragraph of section 3.8 Candidate Drugs for Race-Specific Treatment of Black Endometrial Cancer Patients” as pasted below:

There are multiple drugs targeted at HER2/NEU, including the humanized monoclonal antibody, trastuzuab. A randomized phase 2 trial of adding trastuzumab to carboplatin-paclitaxel showed that this combination was well-tolerated and improved PFS by 4.6 months over chemotherapy alone in patients with advanced-stage or recurrent USCs that overexpress HER2/NEU [89]. This improvement was >8-month for patients with advanced-stage disease undergoing primary treatment, and >3 months for patients with recurrent disease.  Follow-up analysis showed that overall survival was increased by >4 months in the trastuzumab arm with no increase in toxicity [90]. Currently there are two clinical trials of HER2/NEU inhibitors for endometrial cancer listed on clinicaltrials.gov. 

Reviewer 2 Report

This is an intersting and intriguing paper: the Authors discus about genetic  differences between white and black patients affected by endometrial cancers. They , using literature data, focalized differences of biological markers for target therapies based on ethnicities.

This is a new and peculiar approach to read personalized medicine that could help to open knowledge.

This is an invitation to address target therapy research with a new perspective

I Think that this review could be useful to spread this poin of view

Author Response

We appreciate the reviewers positive comments.